# No-Boundary Wave Functional and Own Mass of the Universe

Natalia Gorobey [1,†], Alexander Lukyanenko [1,†] and Alexander V. Goltsev [2,*,†]

1   Department of Physics, Physical-Mechanical Institute, Peter the Great Saint Petersburg Polytechnic University, Polytekhnicheskaya 29, 195251 St. Petersburg, Russia; alex.lukyan@mail.ru (A.L.)
2   Ioffe Physical-Technical Institute, Polytekhnicheskaya 26, 195251 St. Petersburg, Russia
*   Correspondence: goltsev@ua.pt
†   These authors contributed equally to this work.

**Abstract:** An alternative formulation of the no-boundary initial state of the universe in the Euclidean quantum theory of gravity is proposed. Unlike the no-boundary Hartle–Hawking wave function, in which time appears together with macroscopic space–time in the semiclassical approximation, in the proposed formalism, time is present from the very beginning on an equal footing with spatial coordinates. The main element of the formalism is the wave functional, which is defined based on the world histories of the universe. This ensures formal 4D covariance of the theory. The wave functional is defined independently of the wave function as an eigenvector of the action operator. The shape of the Origin region, together with the boundary conditions, is determined by the structure of the total energy of the universe, which includes a 3D-invariant contribution of the expansion energy. The own mass of the universe arises as a non-zero value of the expansion energy in the Origin.

**Keywords:** universe; time; own mass; quantum; Euclidean instanton

## 1. Introduction

The question of the origin of the universe has been and remains central to cosmology. In this work, we will focus on the idea of the quantum birth of the universe from "nothing" [1–6]. This theory was most consistently developed within the framework of the Euclidean quantum theory of gravity (QTG) in the works of Hartle, Hawking, and Hertog [7]. The main object in this approach is the representation of the no-boundary wave function of the universe in the form of a functional integral

$$\psi = \int \prod J dg d\varphi \exp\left(-\frac{1}{\hbar}\widetilde{I}_{GR}\right), \tag{1}$$

where $\widetilde{I}_{GR}$ is the action of General Relativity in Euclidean signature; see also [8]. Integration is carried out over all Euclidean 4D metrics and configurations of matter fields with given values on a single 3D boundary, and $J$ is the Faddeev–Popov determinant. However, in practice, when using polar coordinates in the Origin [7], integral Equation (1) is considered as a representation of the Green's function for the Wheeler–De Witt (WDW) equation with two boundary surfaces, one of which is contracted to a point—a pole. In this case, it is not possible to completely get rid of the boundary conditions for the fundamental dynamic variables at the pole. In particular, the initial value of the scalar field remains a free parameter [7]. A more aggravating circumstance is the fact that integral Equation (1) diverges and the no-boundary wave function can be given meaning only within the framework of the semiclassical approximation. Therefore, in subsequent work [9], the authors considered it reasonable to state the problem in the semiclassical approximation without using a functional integral, directly for the WDW equation, or through the holographic principle [10]. The reason for the divergence of integral Equation (1) is the uncertainty of the sign of the Hilbert–Einstein action. This problem is closely related to the problem of

the positivity of the gravitational field energy [11]. The latter was solved thanks to the proof of the positive energy theorem for the case of asymptotically flat geometry [12,13]. A modification of this theorem for the case of a closed universe is considered in [14]. Here, there is an irremovable negative contribution to energy, which is entirely related to the expansion of the universe.

This paper proposes a formalism alternative to the functional integral Equation (1) on the basis of the invariant wave functional $\Psi[g(x,t), \varphi(x,t)]$, which is defined based on the space of 4D world histories of the universe. To avoid terminological confusion, we immediately emphasize that the wave function $\psi\left(g_{ik}(x), \varphi(x), N, N^k, t\right)$ is a functional of the functions $g_{ik}(x), \varphi(x)$ on a 3D spatial section at a given time $t$ and a functional of the given lapse and shift functions $N, N^k$ [15]. To determine the wave functional, the work [16] formulated the quantum principle of least action, according to which the wave functional is an eigenvector of the action operator.

In the new formalism, the integration over $N, N^k$ is initially absent. In the covariant quantum theory, based on the Batalin–Fradkin–Vilkovysky theorem [17,18], the integration over the lapse function $N$ is equivalent to the integration over proper time (see [19]), so in the new formalism, time remains a free parameter. This makes it possible to formulate a boundary value problem for the wave functional in the "subpolar" region (Euclidean instanton), in which the pole is an internal point, without any additional conditions for the fundamental dynamic variables in it. To fix time in an instanton, one additional parameter will be required—the own mass of the universe.

The next section formulates the basic concepts of the canonical formalism and a new description of the dynamics in the quantum theory of gravity. The second section gives a representation of the energy of a closed universe using spin variables. In the third section, the boundary value problem for the Euclidean instanton is considered in the case of a homogeneous isotropic model of the universe, in which the concept of its own mass arises. In the fourth section, a new canonical representation of the action of the theory of gravity is introduced, based on the energy structure of a closed universe, in which the own mass is realized in the form of a mass spectrum of individual 3D-invariant dynamic modes.

## 2. Wave Functional in the Quantum Theory of Gravity

Let us start our consideration with the classical action of general relativity

$$I_{GR} = -\frac{1}{16\pi G}\int \sqrt{-g}d^4xR + I_m[g,\varphi]. \tag{2}$$

Using $3+1$ splitting of the metric

$$ds^2 = (Ndt)^2 - g_{ik}\left(dx^i + N^idt\right)\left(dx^k + N^kdt\right), \tag{3}$$

let us write it in the canonical form of Arnovitt, Deser, and Misner (ADM) [20]:

$$I_{ADM} = \int dt \int_{\Sigma} d^3x \left(\dot{g}_{ik}\pi^{ik} - N\mathcal{H} - N_i\mathcal{H}^i\right), \tag{4}$$

$N_i = g_{ik}N^k$ , where

$$\mathcal{H}\left(\pi^{ik}, g_{ik}, \pi_{\varphi}, \varphi\right) = -\frac{1}{\sqrt{g}}\left[Tr\pi^2 - (Tr\pi)^2\right] + \sqrt{g}R, +\mathcal{H}_m, \tag{5}$$

$$\mathcal{H}^i\left(\pi^{ik}, g_{ik}, \pi_{\varphi}, \varphi\right) = 2\pi^{ik}_{|k} + \mathcal{H}^i_m \tag{6}$$

are Hamiltonian and momentum constraints and the canonical momenta conjugated to the 3D metric tensor $g_{ik}$ have the form

$$\pi^{ik} = \sqrt{g^{(3)}} \left( g^{ik} Tr\mathbf{K} - K^{ik} \right), \tag{7}$$

$$K_{ik} = \frac{1}{2N} \left( N_{i|k} + N_{k|i} - \frac{\partial g_{ik}}{\partial t} \right). \tag{8}$$

The last terms in Equations (5) and (6) are the energy and momentum density of the matter fields, respectively.

In order to describe the evolution of the universe in QTG in terms of world histories, we introduce the state functional $\Psi$. We define it as the product of wave functions $\psi \left( g_{ik}(x), \varphi(x), N, N^k, t \right)$ on spatial sections $\Sigma_n$ for each time $t_n = \varepsilon n, \varepsilon = T/n$. We suppose that the time dependence of the wave function is determined by the Schrödinger equation

$$i\hbar \frac{\partial \psi}{\partial t} = \int_\Sigma d^3 x \left( N\widehat{\mathcal{H}} + N_k \widehat{\mathcal{H}}^k \right) \psi. \tag{9}$$

Consequently, the wave function $\psi$ is also a functional of $N, N_k$, and the WDW wave equations

$$\widehat{\mathcal{H}}\psi = \widehat{\mathcal{H}}^i \psi = 0 \tag{10}$$

are not initially postulated in our approach, which means they may not be fulfilled. For the wave functional $\Psi$, the normalization condition is assumed to be satisfied:

$$\langle \Psi | \Psi \rangle = \int \prod J dg d\varphi \overline{\Psi}\Psi. \tag{11}$$

It should be assumed that, being a functional of 4D geometry (including the lapse and shift functions $N, N_k$), the wave functional is an invariant of general covariant transformations. The assumption is based on the fact that the basic equation of motion—the Schrödinger equation Equation (9)—for the wave function $\psi$ can be equivalently replaced by the corresponding equation for the wave functional $\Psi$. The latter is a secular equation for the action operator, which is obtained by directly quantizing the action of ADM Equation (4) [16]. This means that we have the opportunity to calculate, for example, the average values of expressions containing the first and second derivatives with respect to time, in particular,

$$\langle \Psi | R_{\mu\nu} | \Psi \rangle, \tag{12}$$

where $R_{\mu\nu}$ is the 4D Ricci tensor. Based on the above, we should expect that expression Equation (12) forms a tensor of the second rank with respect to arbitrary transformations of space–time coordinates, as in the classical theory. This follows from the fact that it is an eigenvector of the action operator. The action operator contains, in particular, the following contribution:

$$\int_\Sigma d^3 x \int_0^T N dt \left[ \cdots + 2\widehat{\pi}^{ik} \frac{1}{2N} \left( \frac{\partial g_{ik}}{\partial t} - N_{i|k} - N_{k|i} \right) + \ldots \right], \tag{13}$$

where $\widehat{\pi}^{ik}$ is the momentum operator, i.e., derivatives with respect to coordinate time and spatial coordinates (together with the lapse and shift functions $N$ and $N^m$) "gathered" into an expression equal to the tensor of the external curvature of the hypersurface $\Sigma$, as was the case in classical general relativity. Since the quantum principle of least action formulated in previous works is equivalent to the Schrödinger equation, we conclude that the latter is also fine with respect to covariance. Formally, this means that arbitrary transformations of time and spatial coordinates, with corresponding transformations of the lapse and shift functions $N$ and $N^m$, provide the necessary transformation properties of all observables. The quantum principle of least action will allow us to determine the structure

of space–time at the beginning of the universe without a priori conditions in the form of the WDW equations. Let us also pay attention to another formulation of dynamics in terms of Heisenberg's operator formalism [21].

### 3. The Energy of a Closed Universe

The lapse and shift functions $N, N_k$ in the new formalism remain arbitrary. Their integration is carried out only under the normalization condition Equation (11). Next, we will introduce a special spin parametrization of these functions, and at the same time, the Ashtekar [22] complex representation of canonical variables of the gravitational field ($\widetilde{\sigma}^k_{AB}$, $A_{KAB}$, $A, B = 0, 1$—spin indices). We immediately take into account the so-called reality condition for the Ashtekar connection, setting

$$A_{kAB} = \Gamma_{kAB}(\sigma) + \frac{i}{\sqrt{2}} M_{kAB}, \tag{14}$$

where $\Gamma_{kAB}(\sigma)$ are components of the real spin-connection, and $M_{kAB}$ are the canonical momenta conjugated to the spin variables $\widetilde{\sigma}^k_{AB}$ in the real representation, in which we can also immediately put

$$M_{kAB} = \frac{\pi_{kl}\sigma^l_{AB}}{\sqrt{g^{(3)}}} \tag{15}$$

(Gaussian constraint $\mathcal{P}^{AB}$ of Ashtekar). Let us introduce the 3D Dirac operator on a spatial section $\Sigma$:

$$\mathcal{D}\eta \equiv i\sqrt{2}\begin{pmatrix} n^A_{A'}\overline{\sigma}^{kA'}_{B'}\overline{\nabla}_k\overline{\mu}^{B'} \\ n^{A'}_A\sigma^{kA}_B\nabla_k\lambda^B \end{pmatrix}, \tag{16}$$

where $\eta$ is the bispinor Dirac field on the spatial section $\Sigma$,

$$\eta = \begin{pmatrix} \lambda^A \\ \overline{\mu}^{A'} \end{pmatrix}, \tag{17}$$

and $n^A_{A'}$ is an arbitrary unitary matrix (spin-tensor) in the spin space. The complex covariant derivative of a spinor field is defined as follows:

$$\nabla_k\lambda_A \equiv \partial_k\lambda_A + A^B_{kA}\lambda_B. \tag{18}$$

Let us introduce anti-involution in the spin space,

$$\lambda^+_A \equiv \sqrt{2}n^A_{A'}\overline{\lambda}^{A'}, \left(\lambda^{++}_A = -\lambda_A\right). \tag{19}$$

We assume that $\sigma^{k+}_{AB} = \sigma^k_{AB}$. Let us also introduce the Hermitian scalar product in the spin space:

$$(\eta_1, \eta_2) \equiv \int_\Sigma \sqrt{g^{(3)}}d^3x\, n_{AA'}\left(\lambda^A_1\overline{\lambda}^{A'}_2 + \overline{\mu}^{A'}_1\mu^A_2\right). \tag{20}$$

It is easy to verify that the Dirac operator Equation (16) is Hermitian with respect to this scalar product. Our constructions are based on the Witten identity, which relates the difference of two positive definite quadratic forms of the bispinor $\eta$ with a linear combination of gravitational constraints in the Ashtekar representation (see [16]),

$$\begin{aligned} (\eta, W\eta) &\equiv -\frac{11}{9}\left(\eta, \mathcal{D}^2\eta\right) + (\eta, (-\Delta + \mathcal{T})\eta) \\ &\equiv \mathcal{L}\left(\widetilde{C}, \widetilde{C}_k, \widetilde{\mathcal{P}}^{AB}\right), \end{aligned} \tag{21}$$

The coefficients of the linear combination are the lapse and shift functions, as well as the zero components of the Ashtekar connection of the form [23]:

$$N = \frac{1}{8} n_{AA'} \left( \lambda^A \overline{\lambda}^{A'} + \overline{\mu}^{A'} \mu^A \right),$$
(22)

$$N^k = -\frac{i}{4} \sigma^k_{AB} \left( \lambda^A \lambda^{+B} + \mu^{+A} \mu^B \right),$$
(23)

$$A_{0AB} = -\frac{1}{16\sqrt{2}} \sigma^m_{(A|C|} \left( \nabla_m \lambda_{B)} \lambda^{+C} + \nabla_m \mu_{B)} \mu^{+C} \right).$$
(24)

The second term in Equation (21) has the form

$$\begin{aligned}
&(\eta, (-\Delta + \mathcal{T})\eta) \\
&= \frac{1}{2} \int_\Sigma \sqrt{g^{(3)}} d^3 x\, n_{AA'} n_{MM'} n_{NN'} \left( \zeta^{AMN} \overline{\zeta}^{A'M'N'} \right. \\
&\quad \left. + \chi^{AMN} \overline{\chi}^{A'M'N'} \right) + (\eta, \mathcal{T}\eta),
\end{aligned}$$
(25)

where

$$\chi^{MNA} \equiv \sigma^{mMN} \nabla_m \mu^A + \frac{2}{3} \epsilon^{A(M} \sigma^{m\,N)}_P \nabla_m \mu^P,$$
(26)

$$\zeta^{MNA} \equiv \sigma^{mMN} \nabla_m \lambda^A + \frac{2}{3} \epsilon^{A(M} \sigma^{m\,N)}_P \nabla_m \lambda \mu^P,$$
(27)

where $\epsilon^{AB}$ is a completely antisymmetric unit spin tensor. Spin tensors Equations (26) and (27) are completely symmetric. The last term on the right side of Equation (25) is a positive definite form of the energy–momentum tensor of matter fields. Thus, identity Equation (21) gives a representation of the Hamilton function of the theory of gravity (right-hand side of Equation (21)) as the difference of two positive definite quadratic forms of the bispinor $\eta$. The fact that we thus obtain the Hamilton function in an arbitrary gauge follows from counting the number of real constraints of the theory of gravity (seven pieces) and the number of independent real parameters of the bispinor $\eta$ (eight pieces). The presence of a redundant parameter leads to the degeneracy of the quadratic form of the operator

$$W = -\frac{11}{9} \mathcal{D}^2 + (-\Delta + \mathcal{T}),$$
(28)

i.e., the existence of a zero eigenvalue for this operator.

In the representation of the Hamilton function of a closed universe Equation (21), separation of the contributions of energy components with different signs has been achieved. The quadratic form $(\eta, \mathcal{D}^2 \eta)$ contains the kinetic energy $(Tr\pi)^2$ (together with the corresponding potential energy), it describes the dynamics of the 3D geometry scale factor $\sqrt{g^{(3)}}$. Therefore, we will call it the energy of space. The quadratic form $(\eta, \Delta\eta)$ does not contain $(Tr\pi)^2$, and describes the dynamics of the "transverse" components of the gravitational field that describe gravitational waves. We will call this, together with $(\eta, \mathcal{T}\eta)$, the energy of matter. The explicit separation of these two components in Equation (21) is a version of the positive energy (of matter) theorem for the case of a closed universe. The combination of signs in Equation (21) also determines the signature of the configuration space of the theory of gravity (superspace).

We can now discuss the issue of regularizing the convergence of the functional integral representation of the kernel of the evolution operator for the Schrödinger equation Equation (9). For the functional integral to converge, it is necessary that the total energy of the universe have a certain sign. This can be achieved by introducing a variable value $e$ instead of the minus sign in Equation (21), which is equal to $+1$ at the calculation stage. In the previously identified wave function, along with the return to real time, the sign of $e$ should also be changed. At the same time, a natural gauge condition would be to take

the eigenvector of the 3D Dirac operator as the bispinor $\eta$. In this case, one can use the Heisenberg formalism [21] in the case of a closed universe.

## 4. Euclidean Beginning of a Homogeneous Isotropic Model of the Universe

The transition to describing the quantum evolution of the universe in terms of world histories and the wave functional allows us to take a fresh look at the problem of initial data for this evolution. In the classical theory of gravity, the timelines of the universe begin at one point, which is the Big Bang singularity. In Euclidean QTG, these lines simply serve as meridians of the "polar" coordinate system [7]. The pole itself has no features other than a coordinate singularity. Therefore, in [24], the state of the universe in the "subpolar" region (with one boundary along the "polar" circle) was proposed to be sought in a non-singular coordinate system using the generalized canonical De Donder-Weyl formalism. And although to introduce time, we return to the usual $3 + 1$ ADM splitting of the metric in polar coordinates, at the pole itself, as an equal point, we place not the initial data for the fundamental dynamic variables $(g, \varphi)$, but their distribution in terms of the wave functional $\Psi[g, \varphi]$. In this sense, we refer to the wave functional of the universe as no-boundary.

Let us consider in more detail the initial stage of evolution of the homogeneous isotropic Friedmann–Lemaitre universe with the metric

$$ds^2 = N^2(t)dt^2 - a^2(t)d\Omega_3^2, \tag{29}$$

where $d\Omega_3^2$ is an element of length on a 3D sphere of unit radius, with a real scalar field and zero cosmological constant. Its dynamics are described by the action (Lorentzian signature)

$$\begin{aligned} I_{FL}[a, \phi] \;=\; & \frac{1}{2}\int_0^T dt \left[ -\frac{a}{\gamma}\left( \frac{\dot{a}^2}{N} - N \right) \right. \\ & \left. + 2\pi^2 a^3 \left( \frac{\dot{\phi}^2}{N} - V(\phi)N \right) \right], \end{aligned} \tag{30}$$

where $\gamma = 2G/3\pi$. The Hamilton function and the corresponding Schrödinger equation for this model are

$$\begin{aligned} h_{FL} \;=\; & N\mathcal{H}_{FL} = N\frac{1}{2}\left[ -\left( \frac{\gamma p_a^2}{a} + a \right) \right. \\ & \left. + \left( \frac{p_\phi^2}{2\pi^2 a^3} + 2\pi^2 a^3 V(\phi) \right) \right], \end{aligned} \tag{31}$$

$$i\hbar\frac{\partial\psi}{\partial s} = \widehat{\mathcal{H}}_{FL}\psi, s = \int_0^t N(t)dt. \tag{32}$$

We will further restrict ourselves to the semiclassical approximation; therefore, we do not consider the problem of ordering noncommuting factors in $\widehat{\mathcal{H}}_{FL}$ here. We also do not consider the problem of convergence of the Euclidean functional integral, which represents the kernel of the evolution operator for equation Equation (32). The extremum conditions for the Euclidean action, which is obtained from Equation (30) after the transition to imaginary time $s = -i\tau$, have the form

$$a\ddot{a} + \frac{1}{2}\left( \dot{a}^2 - 1 \right) - 3\pi^2 a^2\gamma\left( \dot{\phi}^2 + V \right) = 0 \tag{33}$$

is the extremum condition in $a$ and

$$\ddot{\phi} + 3\frac{\dot{a}}{a}\dot{\phi} - \frac{1}{2}V'(\phi) = 0 \tag{34}$$

is the extremum condition with respect to $\phi$, where the dot denotes the derivative with respect to $\tau$, $\tau \in [0, T]$. Let us immediately note that the constraint equation $\mathcal{H}_{FL} = 0$ is not among the extremum conditions, since the lapse function $N$ is not considered as a dynamic variable, and the integral of it is the proper time $s$.

Now, let us consider the problem of boundary conditions for differential Equations (33) and (34). In [7], the Euclidean functional integral of the form Equation (1) is observed in a compact region of 4D Riemannian space with a single boundary on which the values of the scale factor $a(T) = b$ and the scalar field $\phi(T) = \chi$ are given. At the "pole," "natural" initial conditions are chosen

$$a(0) = 0, \dot{\phi}(0) = 0. \tag{35}$$

However, the composition of the equations–extremum conditions in the work [7] differs from that of Equations (33) and (34). Since integral Equation (1) contains additional integration over proper time, the constraint equation also arises under extremum conditions. And since the constraint is also the first integral of the equations of motion Equations (33) and (34), one of them, namely equation Equation (34), can be considered redundant. With this formulation of the boundary value problem, the free parameter turns out to be the value of the scalar field at the pole $\phi(0) = \phi_0$. But this contradicts the very idea of constructing a no-boundary wave function, which assumes the absence of any initial data for fundamental dynamic variables in the polar region. This does not apply to conditions Equation (35), which arise precisely as a result of the choice of a polar coordinate system in a homogeneous isotropic model of the universe.

Let us see how the second of the "natural" conditions, Equation (35), arises if we consider it as the primary representation of the evolution operator in non-singular coordinates in the subpolar region. Moving along the meridian to the pole (one of the timelines in polar coordinates), beyond the pole, we will smoothly continue this movement along the opposite (at an angle $180^0$) meridian, connecting them into one timeline of a non-singular coordinate grid. Let us divide this time axis into small sections of length $\varepsilon$ and write the contribution of the scalar field to the functional integral for the evolution operator of the pole and neighboring points located symmetrically:

$$\int ... d\phi_0 ... \exp\left\{-\frac{1}{\hbar}\pi^2\left[\left(\frac{a_{-1}}{2}\right)^3\right.\right.$$
$$\times\left(\frac{(\phi_0 - \phi_{-1})^2}{\varepsilon} + V\left(\frac{\phi_0 + \phi_{-1}}{2}\right)\varepsilon\right)$$
$$\left.\left.+\left(\frac{a_1}{2}\right)^3\left(\frac{(\phi_0 - \phi_1)^2}{\varepsilon} + V\left(\frac{\phi_0 + \phi_1}{2}\right)\varepsilon\right)\right]\right\} \tag{36}$$

To calculate this integral using the steepest descent method, we find the extremum of the exponent in $\phi_0$, which (in the limit $\varepsilon \to 0$) gives: $\phi_0 = \phi_1$. Here, we also take into account the symmetry of the model under consideration, $\phi_1 = \phi_{-1}$, $a_1 = a_{-1}$. Thus, the second condition in Equation (35) arises as a consequence of estimating the integral over $\phi_0$ in the functional integral representation of the propagator. The presence of this integral also means that the initial condition for the wave function at the pole (at $\tau = 0$) should be taken

$$\psi_0 = A\delta(a). \tag{37}$$

Thus, natural initial conditions mean that initially, $a = 0$, and the field $\phi$ can take on any value with equal probability.

To complete the formulation of the boundary problem, we define the boundary conditions at $\tau = T$. Equations (33) and (34) determine the initial instanton in the Euclidean region if its right boundary point on the $a$-axis is a cusp point, i.e.,

$$\dot{a}(T) = 0. \tag{38}$$

Thus, the history of the scale factor $a(\tau)$ in the instanton is completely determined. For a given $T$, the history of the scalar field $\phi(\tau)$, including its initial $\phi_0$ (as well as final $\phi(T)$) value, also becomes completely determined, since the shape of the potential well for the instanton $a(\tau)$ is determined by the function $\phi(\tau)$. There remains one undefined parameter $T$, fixed by us. We can still calculate the first integral of the equations of motion, which in the general case is constant, but not equal to zero:

$$\mathcal{H}_{FL}(\tau) = -M^2 \neq 0. \tag{39}$$

As we remember, the constraint equation $\mathcal{H}_{FL} = 0$ serves to precisely determine the time of movement $T$ in the generally accepted approach. However, here, this constraint equation, in the presence of a free time parameter, does not follow from anywhere, and we are forced to accept as an additional possibility the presence of a non-zero own mass of the universe $M^2$ in Equation (39). The result can be formulated differently: if the own mass of the universe is given, the shape of the initial instanton in the Euclidean QTG with its own time is completely determined. The minus sign in Equation (39) follows from the analysis of the asymptotic behavior of the scale factor at the pole. It is easy to check that

$$a \sim \left(\frac{9}{2}\right)^{1/3} M^{2/3} \tau^{2/3}$$
$$+ \frac{9}{20 M^{2/3}} \left(\frac{2}{9}\right)^{1/3} \tau^{4/3} + ... \tag{40}$$

at $\tau \to 0$. Thus, the spatial part of the energy of the universe dominates in the beginning, and this serves as a source of its expansion. The simple asymptotic behavior demonstrated in Equation (40) and the entire expansion picture will change if we also take into account the dynamics of anisotropy near the beginning [15]. However, the main term in asymptotics Equation (40) will be preserved, as well as the meaning of the constant $M$. The proper mass remains constant only in a homogeneous isotropic model of the universe. In general, this is not the case, and the dynamics of one's own mass can be directly related to the universe's own time.

## 5. Own Mass and Proper Time in an Inhomogeneous Universe

To establish the connection between proper mass and proper time in the general case, let us consider the new canonical representation of the theory of gravity, which is naturally induced by the representation of the Hamilton function Equation (21). If we consider the bispinor $\eta$ as an independent dynamic variable, then the corresponding Euler–Lagrange equation has the form:

$$W\eta = 0. \tag{41}$$

Taking into account that $\eta$ is initially considered as an arbitrary bi-spinor, we obtain a representation of the system of gravitational connections in the form of an operator equation

$$W = 0. \tag{42}$$

The operator $W$ is Hermitian on the space of bispinors and its spectrum is real. The operator itself is equal to zero if and only if all its eigenvalues $w_n$ are equal to zero. The eigenval-

ues, as well as the eigenvectors $\eta_n$, are functions of the fundamental canonical variables. The eigenvalues $w_n$ form a closed algebra with respect to Poisson brackets:

$$\{w_n, w_m\} = C_{nm}^p w_p, \tag{43}$$

in which the structural "constants" $C_{nm}^p$ are determined by the eigenvectors $\eta_n$, i.e., are also functions of canonical variables. Going forward, we will refer to eigenvalues $w_n$ as dynamic modes. Expanding an arbitrary bispinor $\eta$ over a complete (orthonormal) set of eigenfunctions,

$$\eta = \sum_n \zeta^n \eta_n, \tag{44}$$

we can represent the Hamilton function of gravity theory as a linear combination of a new set of constraints:

$$(\eta, W\eta) = \sum_n L^n w_n, L^n = |\zeta^n|^2. \tag{45}$$

Arbitrary Lagrange multipliers $L^n$ under infinitesimal general covariant transformations generated by $w_n$ constraints,

$$\delta A = \delta s^m \{A, w_m\}, \tag{46}$$

where $A$ is an arbitrary function of canonical variables, must be transformed as follows

$$\delta L^n = \delta \dot{s}^n - C_{mp}^n L^m \delta s^p \tag{47}$$

to ensure action invariance. These infinitesimal transformations are generated by infinitesimal shifts of the proper time parameters $s^n$, and the generators of these shifts are the eigenvalues $w_n$. To determine the Lagrange multipliers corresponding to finite values of the proper time parameters, equation Equation (47) can be solved iteratively, and the solution can be represented as a power series:

$$L^m = \Lambda_n^m(s)\dot{s}^n, \tag{48}$$

$$
\begin{aligned}
\Lambda_n^m(s) \;=\; & \delta_n^m - C_{np}^m s^p \\
& + \frac{1}{2!} C_{rp}^m C_{nq}^r s^p s^q + \ldots.
\end{aligned}
\tag{49}
$$

The proper time parameters introduced in this way are integrals of the Lagrange multipliers:

$$\int_0^T dt L^m(t) = \int_0^{S^n} \Lambda_n^m(s, C) ds^n. \tag{50}$$

The values of the canonical variables in the structure functions $C_{np}^m$ are taken at the same moment of coordinate time $t$ as the proper time parameters $s^p$. The time evolution of the eigenvalues $w_n$ is determined by the equations

$$
\begin{aligned}
\frac{dw_n}{dt} \;=\; & \frac{\partial w_n}{\partial s^p}\dot{s}^p = \{w_n, L^m w_m\} \\
\;=\; & \left\{w_n, \Lambda_p^m\right\}\dot{s}^p + \Lambda_p^m C_{nm}^q w_q \dot{s}^p,
\end{aligned}
\tag{51}
$$

i.e.,

$$\frac{\partial w_n}{\partial s^p} = \left\{w_n, \Lambda_p^m\right\} + \Lambda_p^m C_{nm}^q w_q. \tag{52}$$

In quantum theory, all these relations should be considered in the form of average values in the state described by the wave functional $\Psi$. It follows that if the eigenvalues $w_n$ are zero at the beginning (classical constraints), they always remain so. In this case, we can

talk about preserving the 4D covariance of the theory. If at first there is a non-zero intrinsic mass in some dynamic mode,

$$w_n = -m_n^2 \neq 0, \tag{53}$$

the distribution of own masses over modes will change over time, and this change itself can be considered as a measure of proper time.

Thus, the Euclidean instanton in the general case has the following structure in polar coordinates (radial coordinate—time axis). At the pole (approaching the pole), the approximation of a homogeneous, isotropic model of the universe with a single dynamic mode described by the Hamilton function $\mathcal{H}_{FL}$ is valid. This will happen when choosing polar coordinates in a small neighborhood of any interior point of a smooth manifold. Accordingly, this dynamic mode can be associated with its own mass $M$ as the only parameter of the universe model. The Euclidean "evolution" of the instanton along the radial axes is given by the equation

$$\frac{d}{dt}\sqrt{g^{(3)}} = \left\{ \sqrt{g^{(3)}}, L^m w_m \right\}. \tag{54}$$

We actually have an infinite set of equations (one for each point of the spatial section). The spatial boundary of the Euclidean instanton is determined by the condition that the derivative of $\sqrt{g^{(3)}}$ with respect to time is equal to zero at all spatial points. This provides a system of equations for determining the complete set of proper time parameters at the boundary, and the system of equations Equation (52) allows us to find the resulting distribution of proper mass over modes.

## 6. Conclusions

The generally accepted formulation of the covariant quantum theory of gravity, based on the WDW equations, as well as using the formalism of the invariant functional integral, gives rise to the problem of time (more precisely, its absence). Along with time, the possibility of introducing any additional quantities, in addition to the set of fundamental dynamic variables and associated parameters of the original Lagrangian, is excluded. However, the observed evolution of the universe (or the generally accepted interpretation of observational data) and the idea of the Big Bang as the beginning of this evolution, one way or another, require the introduction of time. This can be achieved by identifying the time parameter with a suitable fundamental dynamic variable [25]. In this case, time acquires a material character in the literal sense of the word, if one of the fields of matter is taken as such a variable. In this paper, an alternative option is proposed—the preservation of the coordinate time parameter of the classical theory of gravity in quantum theory. This is achieved by transition from the description of the quantum state of the universe from a 3D distribution on a spatial section $\Sigma$ to a description in terms of the wave functional on 4D world histories. With this modification, the formal covariance of quantum theory is preserved in the same form as in the classical one, when time and spatial coordinates were equal. However, this equality is actually violated in the case of a closed universe by the signature of the configuration space: the negative contribution in it is clearly highlighted by the 3D-invariant quadratic form of the expansion energy, corresponding to the degrees of freedom of the scale factor $\sqrt{g^{(3)}}$. This energy structure of the universe determines the shape of the initial Euclidean instanton in the semiclassical approximation. This 3D-invariant energy structure is also associated with the spectrum of parameters of the proper time and the canonically conjugate spectrum of parameters of the own mass of the universe. If the proper mass, the distribution and motion in space can be associated with a selected reference frame, which is assumed to be equal to zero, there is no physical reason for the violation of the 4D covariance of the theory. General covariance can be preserved even with a non-zero own mass if it is a constant of motion. But this is possible when the structure constants in Equation (43) are equal to zero, i.e., the dynamic modes in the theory of gravity

are completely independent. This possibility is not excluded, but a detailed analysis of the new canonical representation of the theory is required.

**Author Contributions:** Investigation, N.G. and A.L.; Draft review and editing, N.G., A.L. and A.V.G. All authors have read and agreed to the published version of the manuscript.

**Funding:** This research received no external funding.

**Data Availability Statement:** Data are contained within the article.

**Acknowledgments:** We would like to thank V.A. Franke for the useful discussions.

**Conflicts of Interest:** The authors declare no conflicts of interest.

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
