# Peer review of "No-Boundary Wave Functional and Own Mass of the Universe"

_universe, doi:10.3390/universe10020101_

Round 1
Reviewer 1 Report
Comments and Suggestions for Authors
The authors develop an approach to quantum cosmology presented by them in Ref. [15]; unfortunately I couldn't find a reference to their paper [22]. The aim of their work is to improve the formalism of quantum theory of gravity and to propose a possible solution to the problem of emergence of time in quantum cosmology.
Author Response
Comments and Suggestions of the Reviewer 1
We appreciate to the Reviewer 1 for his/her positive estimation of our work.
Reviewer 1 wrote:
“I couldn't find a reference to their paper [22].”
Our reply:
We thank the reviewer for pointing out our oversight. We have restored the link to our paper [24], see the line (105).
Reviewer 2 Report
Comments and Suggestions for Authors
“No-boundary Wave Functional and Own Mass of the Universe” by Natalia Gorobey Alexander Lukyanenko and A. V. Goltsev
Referee Report
The question of the birth of the universe from “nothing” has a long history. Therefore, the authors’ attempt to approach an old problem from a new angle can be welcomed. Basic works on gravitational instantons are collected in “Euclidean Quantum Gravity” (1993, eds. Gibbons & Hawking). In application to quantum cosmology, the basic idea was to explain the origin of inflationary Universe of Lorentzian signature by tunneling from a Riemannian space of Euclidean signature or from “nothing”. Tryon (1973) was the first to propose that “…our Universe did indeed appear from nowhere” and to mention that “…such an event need not have violated any of the conventional laws of physics”; Zeldovich (1981) who discussed a quantum creation of the Universe; Atkatz & Pagel (1982) who proposed that “…the Universe arises as a result of quantum-mechanical barrier penetration”; “no boundary proposal” by Hartle &Hawking (1983) and birth of inflationary Universe by tunneling from “nothing” by Grishchuk & Zeldovich (1982) , Vilenkin (1982). In the framework of quantum gravity, tunneling from a Riemannian space of Euclidean signature was considered by Gibbons & Hartle (1990).
In the early 80s of the last century, the “fashion” for “generating” the universe by instanton tunneling from Euclidean space to Lorentzian space was based on experimental data of that time, according to which the density of matter in the universe was considered greater than critical, i.e. the Universe seems to have a positive curvature. This fact made it possible to obtain nonsingular solutions (for some sorts of scalar fields) in Lorentzian space (Grishchuk and Zeldovich, 1982; Vilenkin, 1982).
Accordingly, this gave rise to a stream of work on gravitational instantons and the birth of the universe from nothing. When it was later discovered that the universe is flat, i.e. its curvature is close to zero, then attempts to apply old results to new conditions (Hawking and Turok (1998)) were unsuccessful (Vilenkin, 1998, Linde, 1998). Thus, a new attempt (this work) to return to the old problem is of interest. It would probably be useful for the authors to familiarize themselves with the latest work known to me on the birth of the universe from nothing, “Nothing into Something and Vice Versa” (Marochnik, 2023).
From a technical point of view, the work raises questions. The use of a pass integral based on the WDW equation seems doubtful, since, despite its attractiveness, the WDW remains just a hypothesis. In fact, the only mathematically correct work on one-loop quantum gravity is Vereshkov & Marochnik, 2013 (VM, 2013), where in particular it is shown the need to consider the ghost sector, which in the one-loop approximation (and absence of matter) turns one-loop quantum gravity into finite theory in accordance with the t’ Hooft-Veltman theorem (1974). I would recommend that the authors pay attention to the VM, 2013 (especially to sections 6 and 7). In any case, the question of the ghost sector, the need to take it into account or its absence within the framework of the WDW formalism requires discussion, which is absent in the work under consideration.
After eliminating these shortcomings, I would recommend the work for publication.
References
Atkatz D. & Pagels H, 1982, Phys. Rev 25, 2065
Faddeev L.D. & Popov V.N., 1967, Phys. Lett. B25, 30
Gibbons G.W. & Hawking S.W., Eds, “Euclidean Quantum Gravity”, World Scientific, Singapore, 1993
Gibbons G.W. & Hartle B.J., 1990, Phys. Rev. D, 42, 2458
Grishchuk L.P. & Zeldovich Ya. B., 1982, in Quantum Structure of Space and Time, ed. by M. Duff and C. Isham, Cambridge University Press, Cambridge, p. 409; see also Preprint IKI (Space Research Institute, USSR Academy of Sci.) #726, 1982
Hawking S.W. & Turok N., 1998, Phys. Lett. B425, 25-32
Hartle B.J. & Hawking S.W., 1983, Phys. Rev, D28, 2960
G. t’ Hooft & M. Veltman, 1974, Annals of Inst. Henri Poincare, 20, 69.
Linde A., 1998, Phys. Rev, D58, 083514,
Marochnik L., Universe, 2023, 9, 445
Tryon E., 1973, Nature, 246, 396
Vilenkin A., 1982, Phys. Lett. 117B, 25; 1998, arXiv:gr-qc/9812027v1
Vereshkov G.& Marochnik L., 2013, J. Mod. Phys., 4, 285-297; arXiv:1108.4256v2 (2011)
Zeldovich Ya. B., 1981, Soviet. Astron. Lett., 7(5), 322
Author Response
Comments and Suggestions of the Reviewer 2
We appreciate to the Reviewer 2 for his/her positive estimation of our work and the his/her recommendation for publication.
Our reply:
We thank the Reviewer 2 for a detailed selection of articles related to the problem of the birth of the universe from “nothing.” In any case, it will be useful for us to better know the history of this issue. Of interest is a cosmological scenario in which the universe periodically goes through the stage of “nothing” (Marochnik L., Universe, 2023, 9, 445). This perhaps echoes Penrose's recent work. However, in this work we focus on a specific question: how does the conservation of time as a parameter of evolution lead to the emergence of an additional parameter of evolution - the own mass of the universe. Let us note in this regard that we are precisely abandoning the formalism based on the Wheeler-De Witt equation and the Euclidean functional integral. It is based on the Schrödinger equation, which is “unfrozen” if the quantum gravitational constraints are not equal to zero. The representation of the evolution operator in the form of a functional integral does not include integration over the lapse and shift functions of the ADM from the very beginning and therefore the question of gauge conditions and the corresponding contributions of ghosts does not arise. The covariance of this approach is expressed in the invariance of the wave functional Ψ with respect to arbitrary transformations of space-time coordinates. From the reviewer's recommended literature, we added a reference to (Gibbons G.W. & Hawking S.W., Eds., “Euclidean Quantum Gravity,” World Scientific, Singapore, 1993) in the introduction, as well as a reference to the work [Vereshkov G. & Marochnik L., 2013] in the third section. In addition, we find interesting the application of the Heisenberg formalism to the dynamics of a closed universe with the Hamilton function in the form (21), where the components of the bispinor field η serve as the gauge variables. In this case, a natural gauge condition is to take one of the eigenfunctions of the 3D Dirac operator as η.
Reviewer 3 Report
Comments and Suggestions for Authors
Report on No-boundary Wave Functional and Own Mass of the Universe:
The paper proposes a Schrödinger-like Wheeler-Dewitt (WD) equation introducing a time parameter. It claims to have introduced special spin parametrization and Ashtekar’s complex representation of canonical variables of the gravitational field, taking into account the reality condition for the Ashtekar connection (I have not found though). The paper suffers from basic conceptual problem and also is full of errors.
1. WD equation is not postulated, rather it is an outcome of quantization of cosmology. The absence of time is understood as: there is nothing external to the universe. The proposal to mathematically introduce a term on the left hand side of the WD equation by hand as in (9) is mathematically and physically wrong. Since, if and are not constrained to vanish, diffeomorphic invariance and essentially the general covariance, one of the two fundamental postulates of GTR is lost. Authors finally in an example chose , which is again an objection against the diffeomorphic invariance. Indeed, this is a challenge against GTR itself, which has so far been successful in explaining various observations.
This single comment goes against publication of the article. But let me point out several more.
2. In eqn. (5), should be instead.
3. Above eqn. (10) it is assumed that are fixed. But, at the beginning of section 3, it is said that ‘integration over them is carried out only under the normalization condition’. Again in eqn. (32) appears. These statements are contradictory. Further suddenly Ψ appears without clarifying the difference between and ψ and Ψ.
4. Wick rotation in the action involves a pair of operations, such as instead. However, field equations from Lorentzian to the Euclidean are obtained under the single operation .
5. In eqn. (32), a new time parameter ‘s’ appears, and Wick rotation is performed for ‘s’, which is not present in (30). As a result, the authors used three time parameters ( ). This is absolutely crazy.
6. The hat over ‘i’ disappeared in the spelling of Lemaître. However, ‘Friedman Lemaître universe’ stands for the particular model, in which universe is decelerating throughout its evolution. The metric (29) is essentially spatially flat Robertson-Walker metric, which now-a-days is also called spatially flat ‘Friedman-Lemaître-Robertson-Walker metric’ or (spatially flat FLRW metric).
7. Now the silly issues, such as: The metric (29) chosen is spatially flat, while the action (30) and the Hamiltonian (31) are for closed (k = 1) metric. If stands for derivative with respect to s, then (31) is meaningless.
8. Finally, the field equations (33) and (34) are wrong. These would be
Where, prime stands for derivative with respect to the Euclidean time parameter. The equations don’t match even if , since a factor ½ appears with . But, here is a prime variable and cannot be set to a fixed gauge. Further, where is the (0,0) equation of Einstein, which would be different from the energy constrained equation, since ? I have also not found the claimed application of Ashtekar variables anywhere.
I cannot recommend publication of the article in ‘Universe’.

Comments on the Quality of English LanguageNot a big issue.
Author Response
Comments and Suggestions of the Reviewer 3.
We appreciate to the Reviewer 3 for his/her remarks. We improved our paper by removing misprints founded the Reviewer 3. We do not agree with some comments of the Reviewer. Below we give a detailed reply on these comments.
The Reviewer 3 wrote:
- «WD equation is not postulated, rather it is an outcome of quantization of cosmology. The absence of time is understood as: there is nothing external to the universe. The proposal to mathematically introduce a term on the left hand side of the WD equation by hand as in (9) is mathematically and physically wrong. Since, if and are not constrained to vanish, diffeomorphic invariance and essentially the general covariance, one of the two fundamental postulates of GTR is lost. Authors finally in an example chose , which is again an objection against the diffeomorphic invariance. Indeed, this is a challenge against GTR itself, which has so far been successful in explaining various observations.»
Our reply:
First of all, let us doubt that the WD equations can be obtained as a consequence of (Euclidean) quantum cosmology. Apparently, the reviewer is referring to the conditions for the independence of the Gibbons-Hawking partition function in the Euclidean quantum theory of gravity from the succession and shift functions on 3D boundaries, which indeed reduce to the WD equations for the functional integral. However, the functional integral itself, generally speaking, diverges due to the uncertainty of the sign of the action. Its convergence has so far been justified only within the framework of the one-loop approximation. We have included a reference to the seminal work (Gibbons G.W. & Hawking S.W., Eds., “Euclidean Quantum Gravity,” World Scientific, Singapore, 1993) on Euclidean quantum theory of gravity in the introduction. In our opinion, this approach is not the only one possible for the formulation of covariant quantum theory. Formally, covariance means the independence of the physical predictions of the theory from the choice of arbitrary Lagrange multipliers in the Hamiltonian—the succession and shift functions. In this work, we draw attention to the fact that the covariance of quantum theory can be ensured without the requirement that gravitational bonds be equal to zero. Specifically, the rationale for this lies in the discussion of formula (12). An alternative to the WD equations in the proposed formalism is the Schrödinger equation (9) or its equivalent formulation in terms of the quantum principle of least action for the wave functional. We can now talk about the evolution of the universe in time t, which “unfreezes” if the quantum connections are not equal to zero. Covariance in this formulation of quantum cosmology means the invariance of the wave functional with respect to the choice of space-time coordinates on world histories, and not the equality to zero of quantum gravitational constraints. Using Witten's identity (21), the Hamilton function of a closed universe can be represented by a linear combination of the eigenvalues of the operator (28), which are obviously 3D invariant. There are also indications that the eigenvalues commute with each other, which also provides 4D covariance. Thus, we can talk about the formal general covariance of our theory. As for the equality for the Friedmann-Lemaitre universe, diffeomorphic invariance is not violated, since it is trivial in a homogeneous isotropic model. Now we will answer on technical comments, for which we are also grateful to the reviewer.
The Reviewer 3 wrote:
“2. In eqn. (5), should be instead.”
Our reply:
Thank you for your comment. A misprint in formula (5) has been corrected.
The Reviewer 3 wrote:
“3. Above eqn. (10) it is assumed that are fixed. But, at the beginning of section 3, it is said that ‘integration over them is carried out only under the normalization condition’. Again in eqn. (32) appears. These statements are contradictory. Further suddenly Ψ appears without clarifying the difference between and ψ and Ψ.”
Our reply:
To avoid ambiguous interpretation, we have removed part of the text after formula (10). The following and shift functions should be considered everywhere as arbitrary functions of space-time coordinates. The integral of the succession function in (32) determines the proper time parameter s. What is wrong here?The difference between the wave function ψ and the wave functional Ψ is emphasized at the very beginning of the work (34-41). The appearance of any of these quantities should always be clear from the context.
The Reviewer 3 wrote:
“4. Wick rotation in the action involves a pair of operations, such as instead. However, field equations from Lorentzian to the Euclidean are obtained under the single operation .”
Our reply:
It is not clear what A is. If this is an action, then is a consequence of . In fact, in the theory of gravity, the Wick rotation can be achieved in two ways: a), b). The result will be the same. In both cases, if the rotation is carried out in the canonical action of the ADM, the transformation of all canonical momenta should be added to this.
The Reviewer 3 wrote:
“5. In eqn. (32), a new time parameter s appears, and Wick rotation is performed for s , which is not present in (30). As a result, the authors used three time parameters ( ). This is absolutely crazy.”
Our reply:
Here we cannot agree with the reviewer's assessment. Yes, there are three time parameters in the discussion (t,s,τ). Parameter t is coordinate time, equal to spatial coordinates . In the case of a homogeneous model of the universe, it is convenient to replace it with the proper time s, which is defined in (30). The convenience is that the lapse function N(t) is thereby eliminated, and the Wick rotation can be performed in both ways a) N→-iN or b) t→-it with the same result s=-iτ.
The Reviewer 3 wrote:
“6.The hat over ‘i’ disappeared in the spelling of Lemaître. However, ‘Friedman Lemaître universe’ stands for the particular model, in which universe is decelerating throughout its evolution. The metric (29) is essentially spatially flat Robertson-Walker metric, which now-a-days is also called spatially flat ‘Friedman-Lemaître-Robertson-Walker metric’ or (spatially flat FLRW metric).”
Our reply:
This remark is not clear to us. Perhaps we are not fully familiar with the generally accepted designations and names. We can only report once again that in the fourth section we consider a homogeneous isotropic model of a closed universe (minisuperspace) with a real scalar field, in which the spatial section is a 3D sphere.
The Reviewer 3 wrote:
“7. Now the silly issues, such as: The metric (29) chosen is spatially flat, while the action (30) and the Hamiltonian (31) are for closed (k = 1) metric. If stands for derivative with respect to s, then (31) is meaningless.”
Our reply:
Another misunderstanding. Line (110) says: “where dΩ23 is an element of length on a 3D sphere of unit radius.”The second part of the remark is not clear: proper time is introduced in formula (32). Formula (31) gives the Hamilton function for action (30), in which time is coordinate t and there is also a lapse function N.
The Reviewer 3 wrote:
“8. Finally, the field equations (33) and (34) are wrong. These would be
(
Where, prime stands for derivative with respect to the Euclidean time parameter. The equations don’t match even if , since a factor ½ appears with . But, here is a prime variable and cannot be set to a fixed gauge. Further, where is the (0,0) equation of Einstein, which would be different from the energy constrained equation, since ? I have also not found the claimed application of Ashtekar variables anywhere.”
Our reply:
We thank the reviewer for identifying inaccuracies in equations (33), (34). In (33) the factor γ is missing in the last term, and in (34) the factor ½ is also missing in the last term. These typos do not affect the main content and are corrected in the improved text.As for the rest, we are forced to disagree with the reviewer. Before formula (33) it is indicated: s=-iτ, and the dot in the equations means the derivative with respect to τ. Thus, the variable N is implied to be “hidden” in s. It must be assumed that in equations (*) N can also be “hidden”. The fundamental difference between our approach and [7] is that we do not consider N a dynamic variable. Previously, we noted this with the word “fixed” value. This means that functional integration over N is no longer carried out, and the corresponding condition for the extremum of the action over N is not considered. This last condition of the extremum is precisely the constrain equation . Now this condition is gone! However, the quantity is still the integral of motion of equations (33), (34), and this integral may not be equal to zero! With exclamation marks we draw attention to the fact that this is the essence of the work.As for the use of Ashtekar variables, without them we cannot introduce the operator W (28). With the eigenvalues of this operator in section (5) we associate the distribution of eigenmass in the general case (not minisuperspace), according to (53). This allows us to assert that the structure of the own mass of the 3D universe is covariant. An additional analysis is required to prove 4D covariance.

Round 2
Reviewer 2 Report
Comments and Suggestions for Authors
no comments
Author Response
We are grateful to the Reviewer 2 for his/her positive estimation of our work.
Reviewer 3 Report
Comments and Suggestions for Authors
Report 2:
Authors introduced on the left-hand side of equation (9) and write “the WDW wave equations are not initially postulated, which means they may not be fulfilled”. This is absolutely wrong. Authors should first read and understand ‘General Theory of Relativity’ before rebutting. It is known and understood universally that Quantum theory and GTR are mathematically incompatible due to the diffeomorphic invariance of GTR, which is ‘General Covariance’ in disguise. Two fundamental postulates ‘General Covariance’ (field equations are invariant under general coordinate transformation) and ‘Equivalence Principle’ (space-time is locally affine to some approximation, which is equivalent to the notion of parallel transport) are the building blocks of GTR. ‘General Covariance’ essentially leads to the Hamilton and three momenta constraints , which are the and equations of Einstein. This Hamilton constraint equation itself leads to all the field equations in view of Hamilton’s equations and when quantized essentially ends up with the well-known WD eqn., in which time remains absent. This also is the reason for the conflict between GTR and QM. More precisely, while the flow of time is universal and absolute in Q.M., in GTR it is malleable and relative. This is the reason why authors have never seen in the WD equation.
The paper is conceptually wrong and should not be published in the ‘Universe’.

Author Response
We see that there remains a misjudgment of our understanding of the WDW equations.
The Reviwer 3 wrote:
«Authors introduced ?ℏ???? on the left-hand side of equation (9) and write “the WDW wave equations ℋ̂?=0= ℋ̂?? are not initially postulated, which means they may not be fulfilled”. This is absolutely wrong.»
Our reply:
No one (including us) questions the postulate of general covariance in GTR. It is known that arbitrary transformations of space-time coordinates are accompanied by corresponding transformations of the components of the metric tensor (and matter fields), including the lapse and shift functions . It is also known that gravitational constraints serve as canonical generators of infinitesimal generally covariant transformations of all dynamic variables, and they themselves form a closed algebra with respect to Poisson brackets. Now let's move on to quantum theory. In the quantum theory of gravity, based on the canonical representation of the ADM, in the case of an island model with an asymptotically flat geometry at spatial infinity and non-zero total energy (say, in the theory of black holes), the Schrödinger equation is still postulated, and the wave function must also additionally satisfy the constraint equations, which ensures the invariance of observables at infinity (energy, momentum, etc.) from arbitrariness in the choice of space-time coordinates “inside” the island system. Using the Ashtekar gauge [23] for the lapse and shift functions, the problem can be reduced to a single Schrödinger equation with a nonlocal Hamilton function [13]. In the case of a closed universe that interests us, the Hamilton function in the ADM representation is reduced to a linear combination of gravitational connections. This means that in the generally accepted approach to quantum theory, when we take the WDW equations as a basis, the Schrödinger equation becomes redundant. But not at all wrong! Here we come to the point of objecting to the reviewer and pointing out the possibility of an alternative approach to covariant quantum theory.First of all, why are we (and not only us) looking for alternative approaches? We know about the successes of loop quantum gravity. But in the case of a closed model of the universe, as in any approach, the problem of time arises. One of the fields of matter has to be isolated (or added) to count time [25]. It seems to us that this solution to the problem is no better than an explicit violation of covariance.The reviewer writes: «This Hamilton constraint equation itself leads to all the field equations in view of Hamilton’s equations». Yes, gravitational constraints (functions ) generate the entire dynamics of General Relativity, since their linear combination is the Hamiltonian function of a closed universe. But it is not the constraint equations that play this role! Our proposal for solving the problem of time in cosmology is as follows: let us return to the Schrödinger equation as the main dynamic equation (and we do not assume a priori that the constraints are zero). Once again, we are not adding ?ℏ???? to the WDW equations as the reviewer imputed to us. Let's now see how things stand with general covariance. The concern is the arbitrary coordinate time t and arbitrary spatial coordinates present in the Schrödinger equation. But there is no problem here. Central to our work is the statement that the wave functional Ψ (it is defined in the work as the product of wave functions at all subsequent moments of coordinate time) is an invariant of general covariant transformations. This follows from the fact that it is an eigenvector of the action operator. The action operator contains, in particular, the following contribution:
(equation) , (13)
where is the momentum operator, i.e. derivatives with respect to coordinate time and spatial coordinates (together with the lapse and shift functions ) “gathered” into an expression equal to the tensor of the external curvature of the hypersurface Σ, as was the case in classical general relativity. Since the quantum principle of least action formulated in previous works is equivalent to the Schrödinger equation, we conclude that the latter is also fine with respect to covariance. Formally, this means that arbitrary transformations of time and spatial coordinates, with corresponding transformations of the lapse and shift functions , provide the necessary transformation properties of all observables. The quantum principle of least action allows us to determine the structure of space-time at the Beginning of the Universe without a priori conditions in the form of the WDW equations. We apologize for such extended explanations, but we hope they save us from discussing a number of the reviewer's general statements, such as «It is known and understood universally that Quantum theory and GTR are mathematically incompatible due to the diffeomorphic invariance of GTR, which is ‘General Covariance’ in disguise». These statements are not an obstacle for anyone in the search for reconciliation between relativistic and quantum principles. To avoid misunderstandings, we changed the text on page 3 after Eq. (10). Now it is written: “are not initially postulated in our approach.”We gave more detailed explanations on this issue. We added a text on page 3 on lines 64-74 below Eq. (13).
The Reviewer 3 wrote:
“This is the reason why authors have never seen ?ℏ???? in the WD equation.”
Our reply:
We responded to this remark in the text on lines 64-74 where we made a significant addition. As for absolute time in quantum mechanics, it defines rational discussion.
